# Sparse Representing Denoising of Hyperspectral Data for Water Color Remote Sensing

Yulong Guo [1,2], Qingsheng Bi [1,2,*], Yuan Li [3], Chenggong Du [4], Junchang Huang [1,2], Weiqiang Chen [1,2], Lingfei Shi [1,2] and Guangxing Ji [1,2]

1    College of the Resources and Environmental Sciences, Henan Agricultural University, Zhengzhou 450002, China; gyl.zh@henau.edu.cn (Y.G.); huangjc1014@henau.edu.cn (J.H.); chwqgis@henau.edu.cn (W.C.); shilingfei@henau.edu.cn (L.S.); guangxingji@henau.edu.cn (G.J.)
2    Human Engineering Research Center of Land Consolidation and Ecological Restoration, Henan Agricultural University, Zhengzhou 450002, China
3    School of Tourism and Urban & Rural Planning, Zhejiang Gongshang University, Hangzhou 310018, China; liyuan@mail.zjgsu.edu.cn
4    Jiangsu Collaborative Innovation Center of Regional Modern Agriculture & Environmental Protection, Huaiyin Normal University, Huai'an 223000, China; ducg@hytc.edu.cn
*    Correspondence: biqs@henau.edu.cn

**Featured Application: Improving quality of hyperspectral $R_{rs}$ curves and images.**

**Abstract:** Hyperspectral data are important for water color remote sensing. The inevitable noise will devalue its application. In this study, we developed a 1-D denoising method for water hyperspectral data, based on sparse representing. The denoising performance was compared with three commonly used methods in simulated and real datasets. The results indicate that: (1) sparse representing can successfully decompose the hyperspectral water-surface reflectance signal from random noises; (2) the proposed method exhibited better performance compared with the other three methods in different input signal-to-noise ratio (SNR) levels; (3) the proposed method effectively erased abnormal spectral vibrations of field-measured and remote-sensing hyperspectral data; (4) whilst the method is built in 1-D, it can still control the salt-and-pepper noise of PRISMA hyperspectral image. In conclusion, the proposed denoising method can improve the hyperspectral data of an optically complex water body and offer a better data source for the remote monitoring of water color.

**Keywords:** hyperspectral; denoising; water color remote sensing; sparse representing; bio-optical model

## 1. Introduction

Hyperspectral techniques have long been utilized in water color remote sensing [1,2], especially for optically complex water [3–5]. The main controllers of the optical properties are the absorption and scattering of the water color parameters, such as phytoplankton, particulate matter, and colored, dissolved organic matter [6,7]. Different pigments also have varied optical characteristics [8]. Therefore, researchers suggested that to accurately capture the concentration of the water color parameters, the spectrometer needs to contain at least 15 bands at specific wavelength in the visible and near infrared spectral ranges [9,10]. Furthermore, to quantitatively determine the chlorophyll concentration ($C_{chla}$) by remote sensing in systems dominated by suspended sediment, the sensor needs a high spectral resolution (10 to 15 nm band width) at 675 and 705 nm [11]. This determines that the hyperspectral data are urgently needed for the building and refining of water color parameter estimation models [12–15].

However, the accurate hyperspectral measurement of water surface reflectance ($R_{rs}$) is challenging. In field measurements, the above water method is popular because it is stable and easy to use. During the measurement, we need to strictly obey some geometries

to limit sun glitter, avoid instrument-shading problems, and retrieve the $R_{rs}$ [16,17]. In addition, we always collect several spectra (ten in our study) at each sampling station and calculate their average values as the result, in order to control random errors in the measurement. Even though, the occurrence of noise is inevitable. Focusing on this problem, the researchers tried to utilize denoising methods on the $R_{rs}$ curves to improve its quality.

These methods can be categorized into three types: 1-D, 2-D, and 3-D methods. The 1-D methods are based only on the spectral, such as kernel regression smoothing (KRS) [18,19], Savitzky–Golay polynomial smoothing (SG) [20], and mean filter smoothing (Mean) [20]. How to choose a proper denoising level is a problem in the 1-D methods, to avoid over-denoising because the spatial information—that can offer extra constraints—is not involved. The 2-D [21,22] and 3-D [23–25] methods are focusing on hyperspectral images. More challenging than field measuring, the hyperspectral image needs to separate the radiated signal from the field of view (FOV) into hundreds of spectra bands in each pixel. Furthermore, the integration time of the sensor is limited under its high orbiting speed. For the hyperspectral remote sensors, achieving high SNR images is more difficult. Therefore, signal processing methods, such as classic principal component analysis [26,27], wavelet transform [21,28,29], and total variation method [30,31], are widely used. Recently developed sparse representing [32,33], low rank [23,34], and deep learning algorithms [35] also showed great potential in hyperspectral data denoising. By comprehensively utilizing the spectral and spatial information, these methods can better retrieve useful signals in the hyperspectral image from the noisy data. Both the 2-D and 3-D methods require spatial information. This means that these methods were limited to denoise hyperspectral images. Thus, in this study, we focus on 1-D methods, which can be applied to both field-measured and remotely sensed hyperspectral $R_{rs}$ data.

Different optical properties will produce a variety of $R_{rs}$ features [36,37], which bring more challenges to the denoising. In this study, in order to keep the universality with high accuracy of the model, we introduced a bio-optical model—a radiative transfer model in the water body—into the sparse-representing algorithm. The method first generates a comprehensive hyperspectral $R_{rs}$ dataset and decomposes the noisy $R_{rs}$ signals into sparse signals and noise signals. By recomposing the sparse signals, the denoising is completed. The yields will provide more accurate data for the optically complex water-color model development and hyperspectral applications.

## 2. Materials and Methods

We collected both field-measured and remotely sensed hyperspectral data in this study. For the field-measured data, the sampling stations were collected from four routes distributed in two coastal provinces of China (Figure 1). In the four routes, we collected 150 hyperspectral curves and water samples in total (Table 1). For the samples collected in Taihu Lake, the $C_{chla}$ were measured in the library. The $C_{chla}$ data were used to evaluated the effectiveness of the denoising method through some popular spectra indices. In addition, we collected a PRISMA hyperspectral image that covered part of Taihu Lake.

**Table 1.** Basic information of the four cruises. OWT represents optical water type. The detailed description of OWTs can be found in Section 2.2.1.

| Sampling Station | Hyperspectral Sample Number | $C_{chla}$ |
|---|---|---|
| Taihu Lake (1 August 2019) | 60 (OWT5: 3, OWT11: 37, OWT12: 20) | √ |
| Hongze Lake (12 November 2020) | 29 (OWT4: 1, OWT5: 14, OWT11: 13, OWT12: 2) | - |
| Qiandao Lake (1 December 2021) | 10 (OWT2: 2, OWT3: 5, OWT9:2, OWT12: 1) | - |
| Hangzhou Bay (26 July 2017) | 51 (OWT5: 17, OWT11: 30, OWT12: 4) | - |

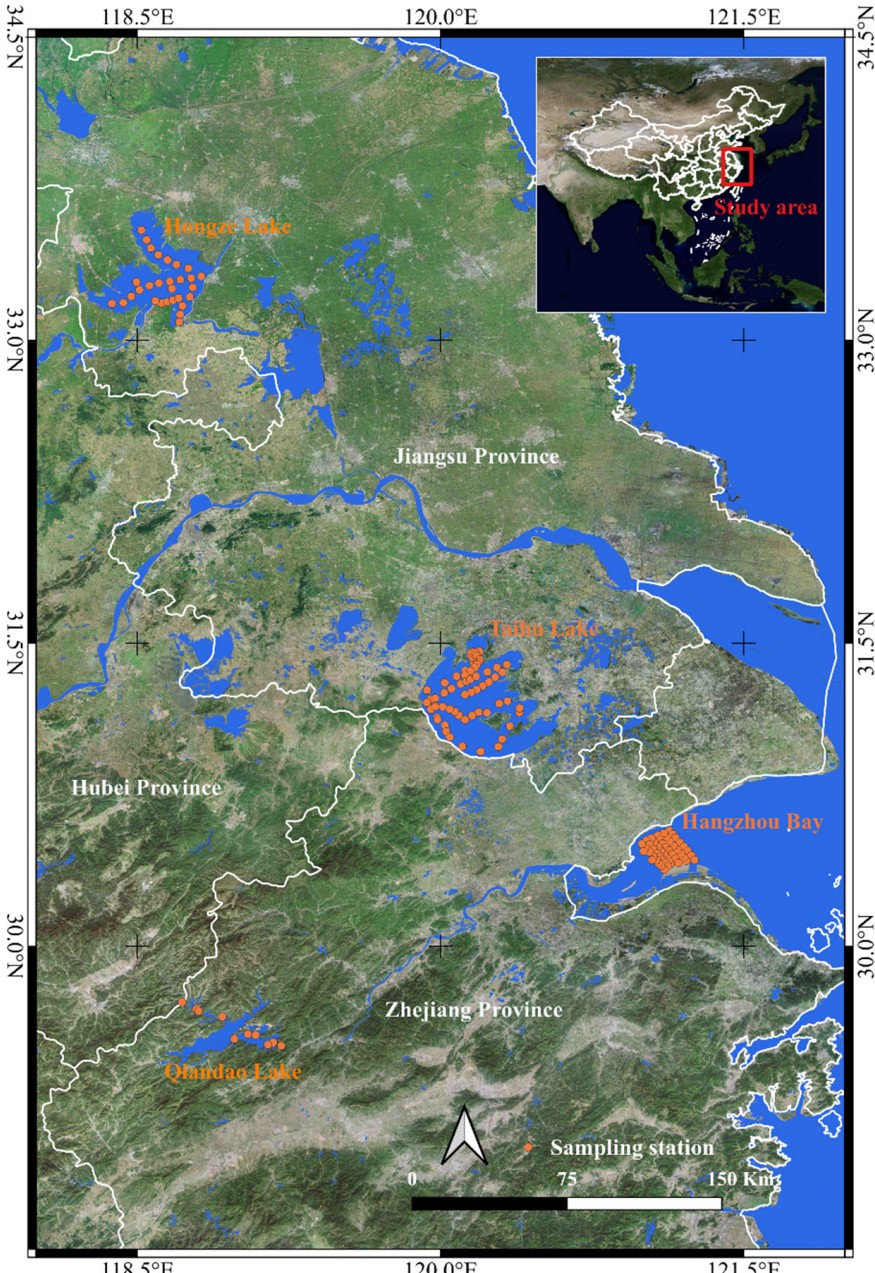

**Figure 1.** Spatial distribution of sampling stations.

*2.1. Study Area*

The field-measured hyperspectral samples were collected in four cruises in Hongze Lake, Taihu Lake, Qiandao Lake, and Hangzhou Bay, respectively. The varied hydrology conditions of the water body provide diversified hyperspectral $R_{rs}$ curves.

Hongze Lake is located in the lower reaches of Huai River in the western Jiangsu Province. It is the junction point of the middle reaches of the Huai River, the tributaries, and the downstream rivers. It plays an important role in the flood regulation of the region [38].

Taihu Lake, located between the Jiangsu and Zhejiang Provinces, is a large shallow lake with an area of ~2338 km$^2$ and a mean depth of ~1.9 m [39]. The lake suffers from water quality deterioration and also shows high turbidity in some areas, due to a large quantity of sediment resuspension. These factors have led to the presence of complex water conditions [40].

Qiandao Lake is the largest freshwater, man-made reservoir in the Yangtze River Delta of China. It is a large (580 km$^2$), deep (mean depth = 30 m), and slightly turbid reservoir that supplies freshwater to more than 10 million residents in the surrounding cities [41].

Hangzhou Bay is the largest bay along the southeastern coast of China, which is located in the northeast Zhejiang Province. It is a wide, shallow, and funnel-shaped estuary with approximately 8500 km$^2$ area, 86 km length, and 95 km wide at the mouth [42]. The water and sediment loads from the Yangtze River into the Hangzhou Bay profoundly influence its hydrodynamics and sedimentation [43].

### 2.2. In Situ Dataset

### 2.2.1. Hyperspectral Data

The in situ $R_{rs}$ was measured using an ASD FieldSpec spectroradiometer, which has a spectral range of 350–1050 nm with increments of 1.5 nm. When the boat was anchored, the radiance spectra of the reference panel, water, and sky were collected, using the above water measurement method [44]. To avoid direct solar radiation and influence of the ship, when measuring the water radiance signals, the azimuth difference of ASD and solar is about 45°. The zenith angle of ASD is also about 45°. When measuring the skylight radiance, the spectroradiometer was rotated upwards by 90–120°. The $R_{rs}$ data for each spectral band were acquired, using the Ocean Optical Protocols:

$$R_{rs}(\lambda) = \left( L_t(\lambda) - rL_{sky}(\lambda) \right) / \left( L_p \pi / \rho_p \right) \tag{1}$$

where $L_t$ is the radiance from water; $L_{sky}$ is the sky radiance; $L_p$ is the radiance measured from a gray reference panel with a diffuse reflectance of $\rho_p = 30\%$; and $r$ is the surface reflectance of the water depending on the wind speed (2.2% for calm weather, 2.5% for <5 ms$^{-1}$ wind, 2.6–2.8% for 10 ms$^{-1}$ wind) [16].

According to Spyrakos' study [45], the inland water spectra can be categorized into 13 OWTs. They also offered the average spectral of each class. The 150 spectra were clustered in terms of the L2 norm distance to the 13 centers. The results are shown in Figure 2. Most of our samples belong to the OWTs 5, 11, and 12, which are sediment-laden waters, waters high in CDOM with cyanobacteria presence and high absorption efficiency by NAP, and turbid, which are moderately productive waters with cyanobacteria presence, respectively. Only a few samples were clustered into the OWTs 2, 3, 4, and 9, which represents the common-case waters with diverse reflectance shape and marginal dominance of pigments and CDOM over inorganic suspended particles, clear waters, turbid waters with high organic content, and optically neighboring to the OWT2 waters but with higher $R_{rs}$ at shorter wavelength. From another aspect, we can see that the samples collected in Hangzhou Bay and Hongze Lake are mainly clustered into the OWTs 5 and 11. The samples from Taihu Lake are mainly belong to the OWTs 11 and 12. Nine out of ten samples collected in Qiandao Lake are categorized to the OWTs 2, 3, and 9. Note that in this research, only the $R_{rs}$ curves between 400 and 800 nm were discussed. This is because the lack of the inherent optical property curves in the wavelength ranges shorter than 400 nm and longer than 800 nm limits the bio-optical simulation.

### 2.2.2. $C_{chla}$ Measurement

The water samples were filtered using GF/C filters (Whatman). The chlorophyll-a was extracted with ethanol (90%) at 80 °C for 6 h in darkness and then analyzed spectrophotometrically at 750 and 665 nm; the phaeopigment correction was carried out using a spectrophotometer (Shimadzu UV-3600, Shimadzu, Kyoto, Japan).

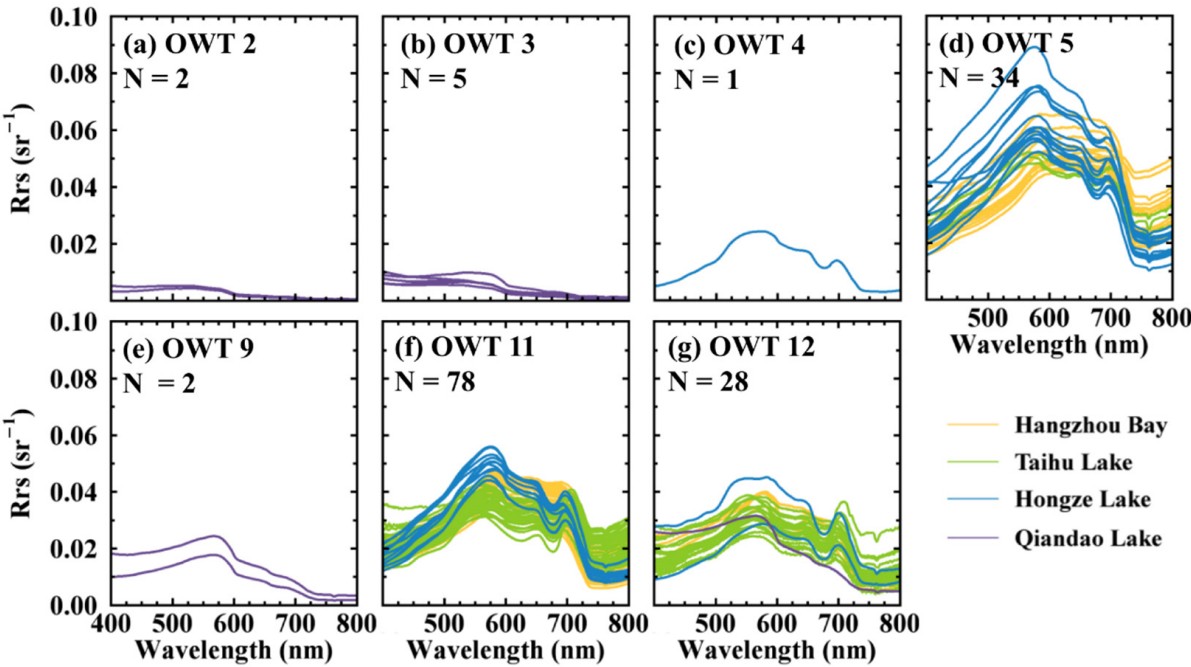

**Figure 2.** In situ measured hyperspectral data that were categorized in OWTs.

### 2.3. PRISMA Image

PRISMA is a small sized satellite mission targeted at qualifying space-borne hyperspectral technology and delivering image spectroscopy data to foster novel processing methods and to employ in a variety of resource management and environmental monitoring applications [46]. It has been successfully applied to remotely estimate water parameters [47]. We collected the PRISMA L2D surface reflectance data on 14 December 2020 to test the performance of our proposed algorithm on the satellite remote-sensing hyperspectral data. The image was rescaled from the original range [0, 65535] to the range [0, 1] and then divided by $\pi$ to obtain $R_{rs}$ data, according to Equation (6).

### 2.4. Denoising Algorithm Description

The proposed sparse representing (SR) method is based on a theoretical hypothesis that an $N$-band hyperspectral $R_{rs}$ curve $\mathbf{R}^{N \times 1}$ is a linear combination of seldom atoms (columns) in a K-column redundant dictionary $\mathbf{D}$. When K $\geq$ N, the target $R_{rs}$ spectrum can be expressed as:

$$\mathbf{R} = \mathbf{D}\alpha \tag{2}$$

where $\alpha$ is the sparse coefficient. $\mathbf{D}$ was generated based on a bio-optical simulated dataset, using the K-singular value decomposition (K-SVD) algorithm. The simulated dataset was generated with wide parameter ranges and contains 10,000 $R_{rs}$ curves. More details of this dataset can be found in [48]. Equation (1) was established based on the assumption that a $R_{rs}$ curve is sparse in a high dimensional linear space ($\mathbf{D}$). When the $R_{rs}$ signal is noisy, Equation (2) should be changed, as follows:

$$\mathbf{R} = \mathbf{D}\alpha + u \tag{3}$$

where $u$ is the random error that is not sparse in the linear space ($\mathbf{D}$). As K > N, the resolution of $\alpha$ is an underdetermined problem. Thus, for an input noisy signal $r$, the sparsity of $\alpha$ can be utilized to solve the problem by:

$$\hat{\alpha} = argmin \|\alpha\|_0 \text{ s.t.} \|r - \mathbf{D}\alpha\|_2^2 = 0 \tag{4}$$



Once the sparsity coefficients were determined, the denoised signal ($\hat{\mathbf{R}}$) can be calculated by:

$$\hat{\mathbf{R}} = \mathbf{D}\hat{\alpha} \tag{5}$$

In addition, we selected three other algorithms as comparisons. They are KRS, GS, and Mean methods.

### 2.5. Assessment Indices

We used *SNR* to evaluate the performance of denoising. It was calculated by

$$SNR = 10lg\left(R_{rs}{}^2 / \left(R_{noisy} - R_{rs}\right)^2\right) \tag{6}$$

where $R_{rs}$ means the reference noiseless signal and $R_{noisy}$ represents the noisy signal.

## 3. Results

### 3.1. Effectiveness of Sparse Representing

The foundation of our proposed method is that each hyperspectral curve can be sparsely represented by predefined atoms. The simulated dataset is used to verify this hypothesis, because it is theoretically noiseless. In detail, we decomposed the simulated dataset using the OMP algorithm and recovered the spectra using the sparse coefficients and the dictionary. The SNR indices between the original spectra and recovered spectra per band are shown in Figure 3. For an ideal recovery, its signal should be the same as original spectral and the SNR value is infinite.

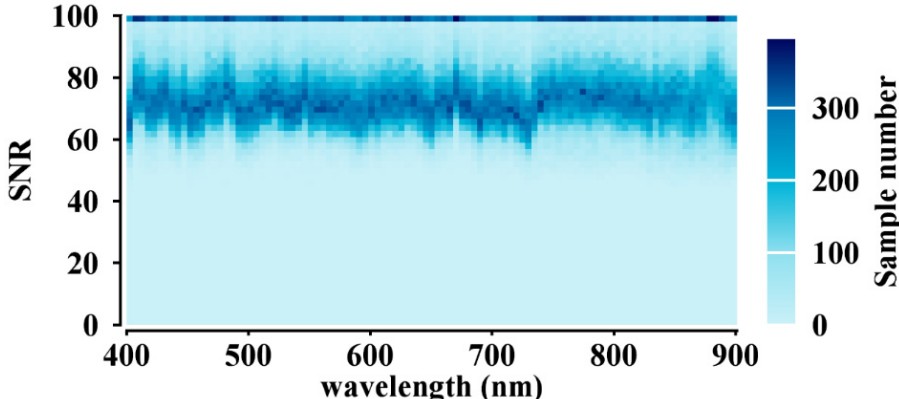

**Figure 3.** Performance of sparse representing method in dictionary dataset.

We calculated the SNR histogram of each of the spectral bands (Figure 3). These histograms have two high density regions. The first one is around the SNR and equals 70 for all of the bands. This means that the proposed method can equally represent all of the spectra in the range of 400 to 900 nm. Two SR-processed spectra with a SNR of 77.41 (Figure 4a) and 67.73 (Figure 4b) indicate that such an error level (the SNR is about 70) is negligible visually. The second high density region that appears in the SNR equals 100. In fact, for these samples, the SR and original signals are numerically indistinguishable. The SNR values are infinite. In order to better show the overall characters of the SNR distribution, we set the maximum SNR as 100 in Figure 3. In conclusion, the proposed algorithm can effectively represent the hyperspectral data of the inland water.

### 3.2. Denoising Performance in the Simulated Dataset

In this section, we added random errors into the simulated dataset used in the previous section, to yield spectra with different SNRs. By controlling the input error, the simulated SNRs resulted in three levels (30, 50, and 70). Then, we applied the SR and the other three



algorithms to the noisy spectra and obtained the denoised data. The SNRs of the spectra before and after denoising were calculated and compared.

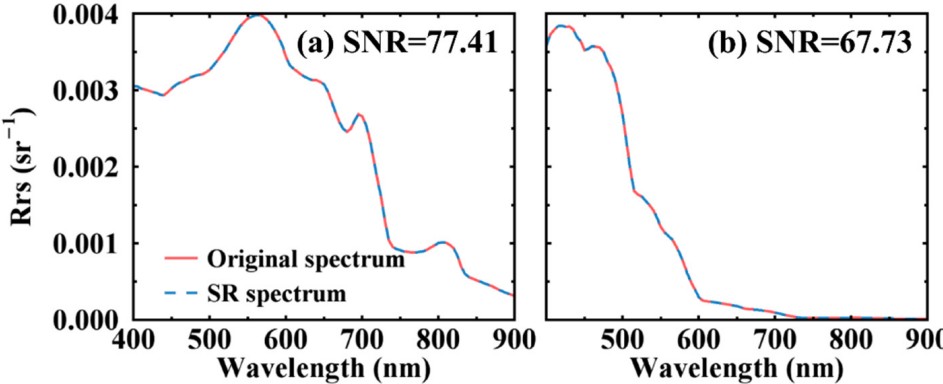

**Figure 4.** Two representative spectra and their corresponding SR spectra. The SNR indices are 77.41 (**a**) and 67.73 (**b**), respectively.

We plotted three typical spectra under varied noise levels, together with the SR denoised ones (Figure 5). When the SNR is around 30, the noisy $R_{rs}$ curves show obvious vibration (Figure 5a). The sharp peaks and valleys caused by random noise at sensitive bands will bring errors into the water-color estimation models. The denoised signals (Figure 5b) look smooth and highly consistent with the original noiseless $R_{rs}$ curves. When the SNR was set to about 50, the noised spectra (Figure 5c) were obviously smoother than those in Figure 5a. We could only see some slight fluctuations in several bands. The SR denoised curves were nearly indistinguishable with the original signals. When the SNR is about 70, both the noisy and denoised spectra looked the same with the original $R_{rs}$ curves. This means that the SR method can yield satisfactory results under both high, median, and low noise levels.

Furthermore, we calculated the SNR of the denoised spectra from the four methods and drew their violin plots. As a comparison, the SNR of the noisy spectra were also calculated (Figure 6). It is interesting that, overall, some of the methods provided even worse results than the noisy signals. In particular, when the SNR is about 70, the SNRs of the denoised spectra are lower than those of the noisy spectra. According to Figure 4, when the SNR was around 70, the noise of the spectra was nearly negligible. Under this situation, all four of the methods destroyed the useful information to different degrees. The SR method kept most of the original signals and its median SNR was 64.30. GS performed better than KRS and Mean; its median SNR was 59.30, which was close to the SR. The SKRS and Mean exhibited a similar performance, their median SNRs were 48.05 and 46.19, respectively. When the input noise level increased, the SNR decreased to about 50 (its median is 52.55) (Figure 6b); the performance of the four methods was similar to that in Figure 6c. What was different is that at this input error level, the SR and GS effectively increased the SNR compared with the noisy signals. Their median SNRs were 56.83 and 53.31, respectively. The KRS and Mean still could not effectively remove the noise and keep useful information synchronously. Their median SNRs were 44.87 and 44.65, respectively. This indicates that they were still over-denoising at this noise level. When we further increased the noise level and the SNR dropped to about 30 (its median is 32.39), the SR-yielded spectra could effectively increase the SNR to a median of 37.49. In the other three algorithms, the Mean outperformed the KRS and GS. Its median SNR was 36.79, which was quite close to the SR. The KRS could also slightly improve the SNR at this input noise level.

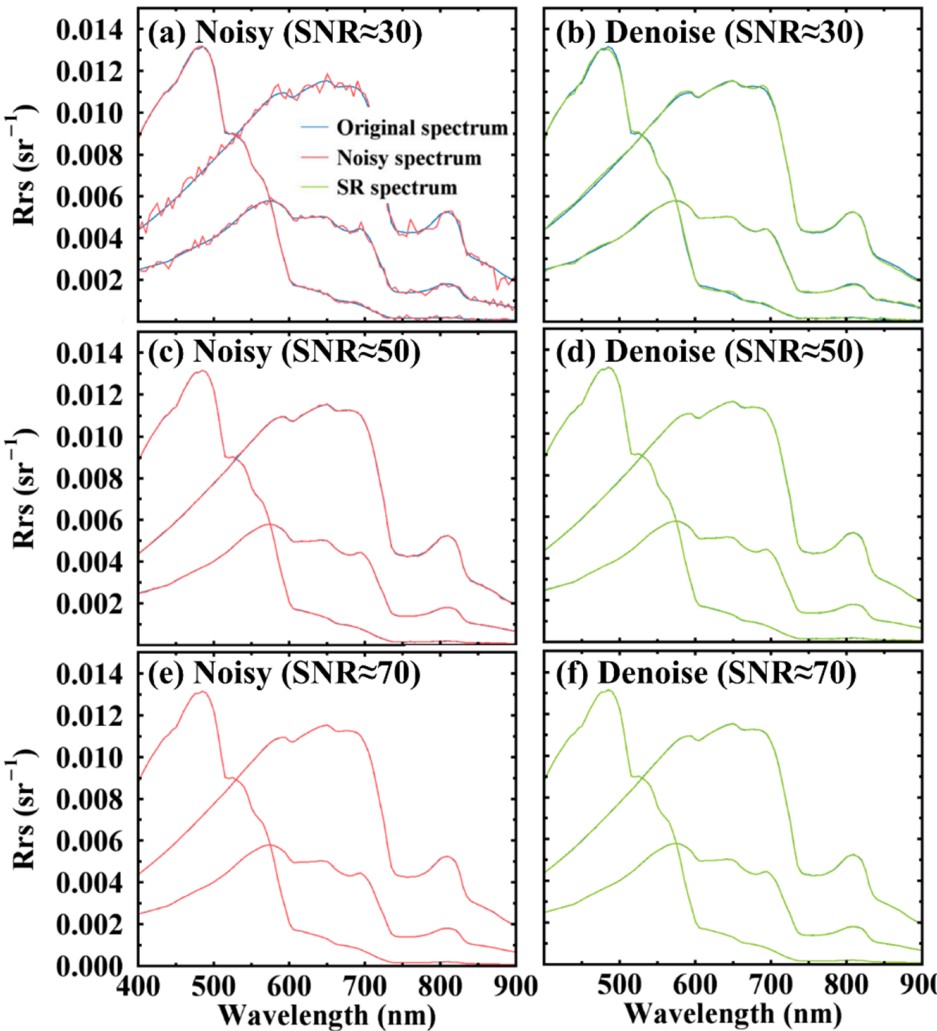

**Figure 5.** Example spectra of three samples under different SNR levels.

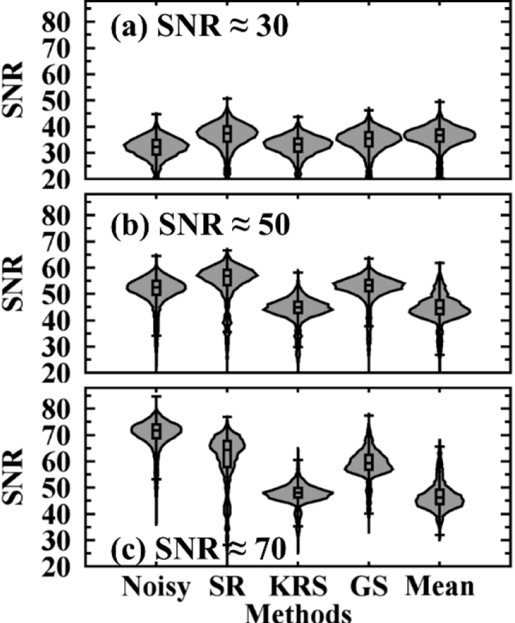

**Figure 6.** Performance of the four methods at different input SNR levels.

### 3.3. Denoising Performance in ASD Measured Dataset

Compared with the remote-sensing image retrieved $R_{rs}$, the field-measured $R_{rs}$ were less influenced by complex absorption and the scattering of gas molecular and particulates in the atmosphere. In addition, the field measurement of $R_{rs}$ had a longer integration time. This will further decrease random errors. So, the field measured spectra are usually treated as the reference data in research, such as atmospheric correction [49–52]. However, the field-measured signal is not noise-free. Random factors, such as wind speed, white cap, and platform swing, will all influence the measurement. Some of the related research completed a pre-process to denoise the $R_{rs}$ curves and then built $C_{chla}$ estimation models. The improved estimation accuracy indicated the necessity of denoising, even for field-measured $R_{rs}$ [18,53].

We selected four typical $R_{rs}$ curves and showed the denoised spectra, together with the original ASD spectra (Figure 7). Different from the simulated noisy spectra (Figure 5), in this dataset, the noises were not evenly distributed along the wavelength. Generally, the spectra curves between 500 and 720 nm are smooth. However, in the shorter and longer wavelength regions, with the decrease in the $R_{rs}$ signals, the noises were more obvious. The denoised curves are highly consistent with the original spectra. All of the characteristic peaks and valleys were clearly kept after denoising; for example, the $R_{rs}$ valley caused by the chlorophyll-a absorption peak at around 680 nm and the peak caused by chlorophyll-a fluorescence near 700 nm. At the same time, the $R_{rs}$ curves at high noise wavelengths became smoother after denoising. The results indicated that, for ASD-measured spectra data, the proposed SR denoising method is also effective. The method is not sensitive to the wavelength distribution of random noise.

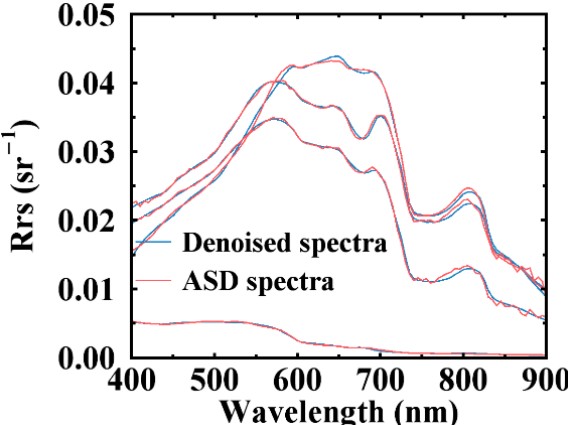

**Figure 7.** Typical ASD measured and their denoised spectra.

### 3.4. Denoising Performance in Hyperspectral Image

The previous sections showed the effectiveness of the SR method in simulated and ASD-measured $R_{rs}$ spectra. In this section, we will show the performance of the SR denoising method on PRISMA hyperspectral data. The lower SNR level of PRISMA (compared with ASD) leads to its stronger vibrations over all of the spectral bands (Figure 8). We can observe two kinds of noises from the spectra. In the wavelength range that is shorter than 550 nm, the noises are relatively wavelength-independent. The PRISMA $R_{rs}$ curves changed randomly along the wavelength. Another kind of noise showed more common characteristics, such as the sharp small peak at around 775 nm. According to the simulated (Figure 5) and field-measured $R_{rs}$ curves (Figure 7), this peak is a systematic error caused by the sensor or the atmosphere. Under this complex error condition, the SR-denoised spectra (Figure 8) look smooth at the random-error region and flatten the small peaks around 775 nm.

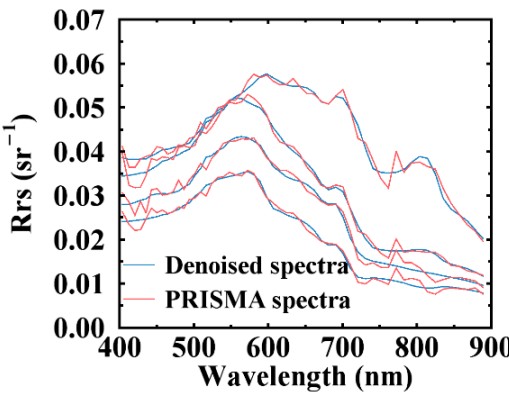

**Figure 8.** Typical PRISMA and their denoised spectra.

In three different datasets, the proposed SR denoising method showed its encouraging performance. This method can handle both random noise at varied levels (Figure 6), unevenly distributed noise (Figure 7), and systematic noise (Figure 8).

## 4. Discussion

### 4.1. Influence of K

Sparsity is an important parameter in our proposed algorithm. It decides how many atoms are involved in compositing the $R_{rs}$ signal. More atoms mean more spectral details are included in the result. Simultaneously, more errors might also be retained. Therefore, we evaluated the performance of the proposed algorithm under different sparsity (k), to give suggestions for the denoising signals at different SNR levels. In the same way as the former sections, all of the calculations were carried out using the simulated dataset at three input SNR levels. To exhibit the results more clearly, we further calculated the SNR difference before and after the denoising. A SNR difference equal to 0 means the denoising made no contribution to the recovery of useful signals. An ideal SNR difference should be larger than 0, which means the denoising is effective. On the contrary, if the SNR difference is less than 0, the denoising destroyed part of the useful signals. The dashed line in Figure 9 represents the SNR difference equals 0.

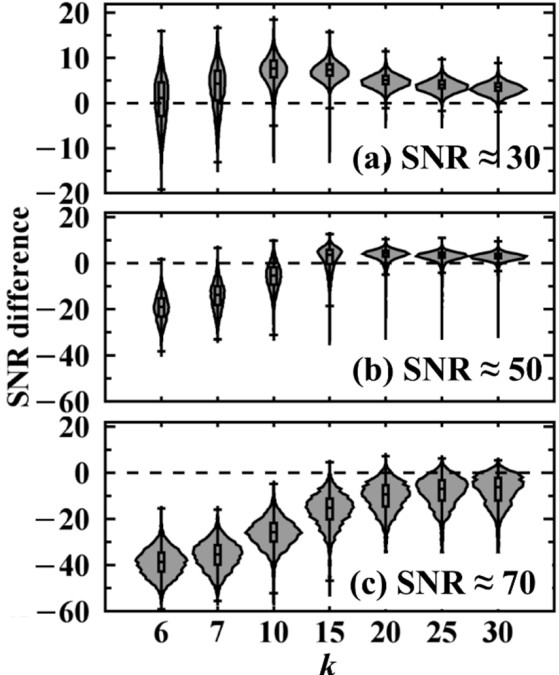

**Figure 9.** SNR difference of the proposed algorithm under different *k* value.

The violin plot of the SNR differences under three input SNR levels is shown in Figure 9. With the increase in k, the denoising performance under different SNR levels exhibited similar trends. When k was less than 10, the SNR differences increased synchronously with k. After k was larger than 10, the SNR differences became flat. This suggests that the SR method is not stable under low sparsity conditions. Another evidence of the stability is the width of the violins in Figure 9a,b. A narrower and longer violin indicates a wider SNR difference distribution. A stable denoising of one dataset is expected to yield a concentrated performance. From this aspect, when effective denoising is generated (Figure 9a,b), a higher sparsity means the model is more stable.

What is interesting is that when the input noise is strong (Figure 9a), the increase in the sparsity will not always lead to a better performance. The SNRs slowly decreased when sparsity was larger than 15, even though they are still generally higher than the dashed line. This denotes that, at highly noisy levels, a larger k value of SR method will keep more of the noise signal. Sparse representation cannot completely separate the noise signal.

Considering both the high and low SNR levels, an optimized sparsity in this research was set to 15. This is also the parameter that was applied in the previous sections. In practice, when given enough prior information, we suggest using a lower sparsity (k = 10) for the low SNR signals (SNR ≈ 30), and a larger one (k = 30) for the high SNR (SNR ≈ 70) signals.

### 4.2. Influence of Denoising to $C_{\text{chla}}$ Estimation Models

The final goal of denoising is the quantitative estimation of the water quality parameters. We take $C_{\text{chla}}$ as the example to evaluate the effectiveness of the SR denoising method in practice. Limited by the amount of the field samples and measurements (Table 1), we cannot refine the estimation models and evaluate the estimation result directly by comparing it with the in situ measured $C_{\text{chla}}$. Therefore, the correlation coefficients between the popular spectral indices and $C_{\text{chla}}$ were selected to complete this assessment.

The five popular spectral indices are the band ratio parameter [54,55], normalized difference chlorophyll-a index (NDCI) [56], three-band parameter [12], four-band parameter [57], and enhanced three-band parameter [58]. Their expressions are listed in Table 2. In our dataset, the three-band index exhibited the highest correlation with $C_{\text{chla}}$. It is followed by the enhanced three-band and band ratio indices. The correlation coefficients of the three-band and four-band indices were slightly lower than the other three indices. For all of the five indices, the denoised $R_{\text{rs}}$ spectra showed better correlation coefficients compared with the original $R_{\text{rs}}$. The improvements are not obvious, because the main wavelengths of the indices are located in the low-noise ranges (Figure 7), where the SR method kept most of the original signals.

**Table 2.** Linear correlation coefficients between $C_{\text{chla}}$ and five spectral indices that calculated from original $R_{\text{rs}}$ and denoised $R_{\text{rs}}$.

| Spectral Index | Expression | Original $r$ | Denoised $r$ |
| --- | --- | --- | --- |
| Band ratio [54,55] | R710/R680 | 0.719 | 0.721 |
| NDCI [56] | (R710 − R680)/(R710 + R680) | 0.697 | 0.700 |
| Three-band [12] | $(1/R680 − 1/R710) \times R745$ | 0.752 | 0.755 |
| Four-band [57] | (1/R680 − 1/R710)/(1/R745 − 1/R720) | 0.695 | 0.698 |
| Enhanced three-band [58] | (1/R680 − 1/R710)/(1/R745 − 1/R710) | 0.743 | 0.748 |

### 4.3. Denoised PRISMA Image

As previously mentioned, the proposed SR denoising method is the 1-D method. Even the denoising is only dependent on the spectra signals; after the method is iterated over each pixel, the denoising effect will affect the spatial information. We tested it on a 3D PRISMA image cube in this section. The original PRISMA image and the denoised image of bands 442 nm, 555 nm, and 700 nm are shown in Figure 10.

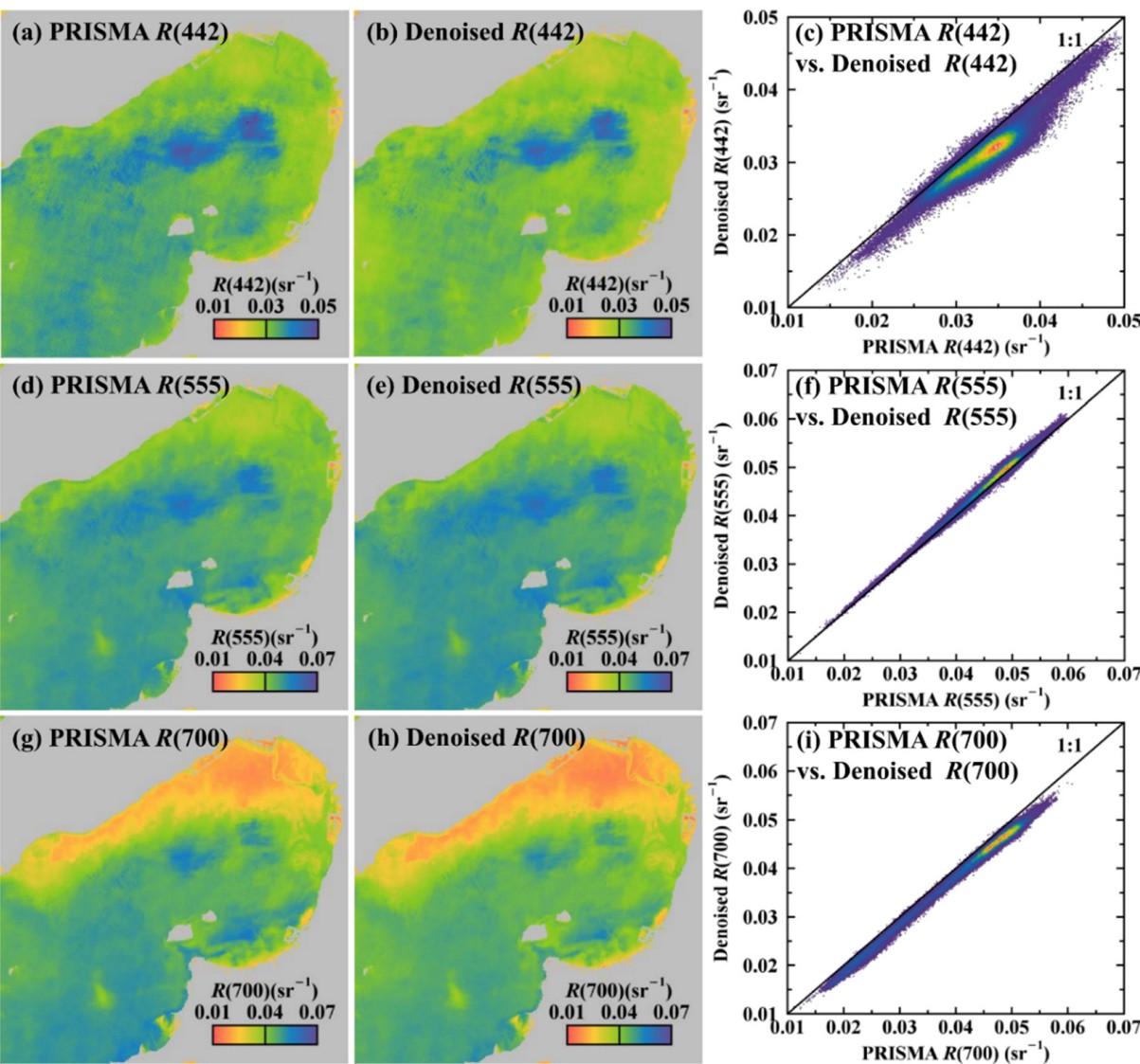

**Figure 10.** PRISMA and SR denoised images of band 442 nm (**a**,**b**); 555 nm (**d**,**e**); and 700 nm (**g**,**h**); and the scatter plots of original and denoised images of (**c**) 442 nm; (**f**) 555 nm; and (**i**) 700 nm.

For each spectral band, the denoising process kept the general $R_{rs}$ spatial distribution of the original image. This further revealed the fidelity of the SR algorithm. In the three bands of the PRISMA image, the $R(442)$ image showed an obvious salt-and-pepper noise. Together with Figure 8, we can infer that the low SNR level of this band caused this error. In other words, the noise that distributed in the spatial and spectral domains are related. Therefore, the denoised image of band 442 (Figure 10b) successfully erased the random errors. For the other two spectral bands, their images look smooth and clear. This means these two bands are at higher SNR levels. The denoising kept most of the original information and slightly changed the reflectance value (Figure 10f,i). The encouraging image validation result suggests that the 1-D SR denoising method effectively improved the quality of the PRISMA image.

Finally, we calculated the $C_{chla}$ maps from the original PRISMA image and the denoised image, using Huang's model [55]. From the results (Figure 11), we can see that, in the same way as with the single band images, the spatial distributions of the PRISMA and the denoised image yielded $C_{chla}$ maps that are similar. Overall, the denoised $C_{chla}$ is highly consistent with the original $C_{chla}$, but systematically lower (Figure 11a,b). The difference between the original and denoised images yielded $C_{chla}$ that are stronger than

the ASD results (Table 2). This is reasonable, considering the lower SNR of the PRISMA data will lead to a larger difference after being denoised (Figure 8). Meanwhile, a higher dynamic range after denoising (Figure 11c) reveals that the SR method enhanced the spatial information-capturing ability of the PRISMA image.

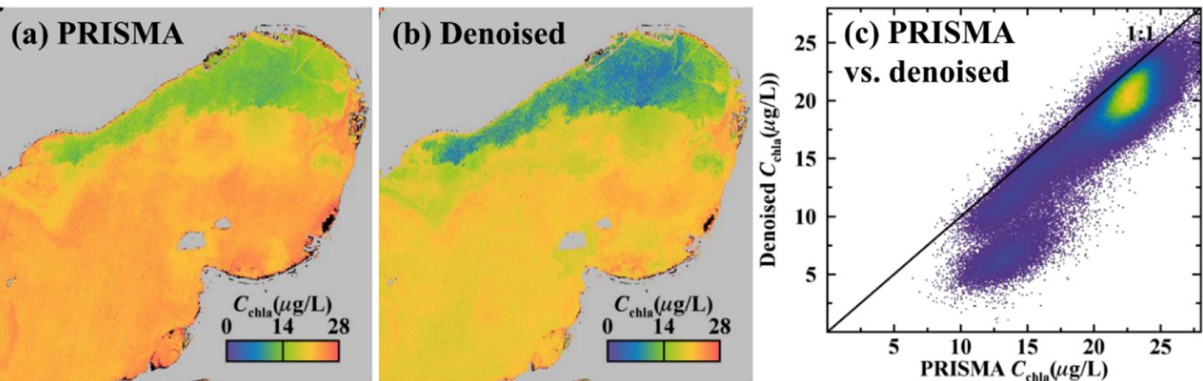

**Figure 11.** $C_{chla}$ maps yielded from (**a**) PRISMA and (**b**) denoised images and (**c**) their scatter plot.

### 4.4. Limitations and Outlooks

A classic problem in signal denoising is its fidelity. How to keep most of the useful signals when effectively wiping out errors is always challenging. We proposed a SR denoising method for optically complex water spectra. Even when it is dependent on the general prior information of the $R_{rs}$ signal—the bio-optical model—it ruined some useful information when the SNR level was high enough (Figures 6 and 9). This means the current algorithm is still improvable in separating the noise from the signal. We believe that, with the help of fast-developing deep learning algorithms, the performance of denoising might be further improved.

In this study, we only focused on hyperspectral data. In practice, multispectral remote sensing data, such as Landsat series image [41,59,60], Sentinel-2 and 3 image [61,62], Moderate Resolution Imaging Spectroradiometer (MODIS) image [63–65], Medium Resolution Imaging Spectrometer (MERIS) image [66–68], Geostationary Ocean Color Imager (GOCI) image [40,69,70], and Gaofen series image [71,72], are more frequently chosen, due to their higher spatial and temporal coverage, and more importantly, their higher SNR that benefits from their limited spectral bands. On the other hand, for these data, the accurate retrieval of the atmosphere parameters, such as aerosol optical depth, water vapor content, and gas content, are more challenging. Therefore, accurate atmospheric correction has long been a problem for the optically complex water color multispectral remote sensing missions. Theoretically, if the atmospheric correction error is randomly distributed along the wavelengths, our proposed SR method can improve the atmospheric correction results. However, for the systematically biased errors, the performance will need a thorough discussion in the future. If possible, the atmospheric-correction-then-denoise scheme will improve the image quality for inland optically complex water monitoring.

### 5. Conclusions

In this research, we developed a SR denoising method for optically complex water hyperspectral $R_{rs}$ data and tested the method in three hyperspectral datasets. The results indicated that the SR method can effectively remove the random errors and systematic errors of the hyperspectral data and keep the useful signals at the same time. The method is robust under different types of error distribution. The denoised hyperspectral data can improve correlation coefficients between the spectra indices and $C_{chla}$ and the dynamic range of $C_{chla}$ maps.

**Author Contributions:** Conceptualization, Y.G. and Q.B.; methodology, Y.G. and Y.L.; validation, Y.G., J.H. and W.C.; formal analysis, L.S. and G.J.; investigation, Y.G., Y.L. and C.D.; data curation, Y.G.; writing—original draft preparation, Y.G.; writing—review and editing, Y.G., Q.B. and Y.L.; visualization, Y.G.; supervision, Q.B.; project administration, Q.B. and W.C.; funding acquisition, J.H., W.C., C.D., Y.L. and G.Y All authors have read and agreed to the published version of the manuscript.

**Funding:** This research was funded by National Key R&D Program of China, grant number 2021YFD 1700900; National Natural Science Foundation of China, grant number 41701422, 42071333, and 42001296; Young backbone teachers program of Henan Province, grant number 2019GGJS048.

**Institutional Review Board Statement:** Not applicable.

**Informed Consent Statement:** Not applicable.

**Data Availability Statement:** Not applicable.

**Conflicts of Interest:** The authors declare no conflict of interest.

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
