# Peer review of "Sparse Representing Denoising of Hyperspectral Data for Water Color Remote Sensing"

_applsci, doi:10.3390/app12157501_

Round 1

Reviewer 1 Report

The authors submitted an interesting manuscript dealing with water color remote sensing using hyperspectral data. However, they should revise it, especially on the design, content, and its structure, before it could be considered for publication. Below are some comments and suggestions to improve the overall quality of the manuscript.

Line 1: Please specify the type of the manuscript.

Lines 29-69: In the section of introduction should provide a brief introduction of their proposed method (universal hyperspectral Rrs denoising method), they should better precise the main aims of this study as well.

Line 70: In the section of Material and Methods, the authors should include the subsection of the Study Area and provide enough description of the studied area. They should provide a subsection containing the description of the dataset. They should provide the subsection of the methodology containing information represented in the lines 82-107.

Lines 134-151: This section should be in the subsection of dataset description and not in the section of Results.  

Lines 32-33: Please provide references of recent studies which have been conducted and dealt with optical water properties. Please refer to (1) El Ouali, A.; El Hafyani, M.; Roubil, A.; Lahrach, A.; Essahlaoui, A.; Hamid, F.E.; Muzirafuti, A.; Paraforos, D.S.; Lanza, S.; Randazzo, G. Modeling and Spatiotemporal Mapping of Water Quality through Remote Sensing Techniques: A Case Study of the Hassan Addakhil Dam. Appl. Sci. 202111, 9297. https://doi.org/10.3390/app11199297, (2) Ahmadi, B.; Gholamalifard, M.; Kutser, T.; Vignudelli, S.; Kostianoy, A. Spatio-Temporal Variability in Bio-Optical Properties of the Southern Caspian Sea: A Historic Analysis of Ocean Color Data. Remote Sens. 202012, 3975. https://doi.org/10.3390/rs12233975.

Line 297: “The five popular spectral indices…” please provide references.

Author Response

Comment: Line 1: Please specify the type of the manuscript.

Response: we have specified the article type in the revised manuscript.

Comment: Lines 29-69: In the section of introduction should provide a brief introduction of their proposed method (universal hyperspectral Rrs denoising method), they should better precise the main aims of this study as well.

Response: Thanks for your valuable suggestion. According to this comment, we rewrote the last paragraph of section 1. In detail:

Line 70. “we build a universal hyperspectral Rrs denoising method based on bio-optical model and sparse representing” was deleted.

Line 67. “Different optical properties will produce varied Rrs features [36,37], which bring more challenges to the denoising. In this study, in order to keep the universality with high accuracy of the model, we introduced bio-optical model­­­­­­ – a radiative transfer model in water body— into the sparse representing algorithm. The method first generates a comprehensive hyperspectral Rrs dataset and decomposes noisy Rrs signals into sparse signals and noise signals. By recompose sparse signals, the denoising completes.” Was added.

Comment: Line 70: In the section of Material and Methods, the authors should include the subsection of the Study Area and provide enough description of the studied area. They should provide a subsection containing the description of the dataset. They should provide the subsection of the methodology containing information represented in the lines 82-107.

Response: Thanks for your suggestion. We have added and reorganized section 2.2 in our revised manuscript. In the current version, section 2.2 contains two subsections: 2.2.1 Hyperspectral data and 2.2.2 Cchla measurement. Section 2.2.1 contains the measurement method of Rrs curves and their characters. And section 2.2.2 contains the measurement of Cchla. In addition, we moved Tab. 1 to the very first paragraph of section 2 to describe the dataset.

Comment: Lines 134-151: This section should be in the subsection of dataset description and not in the section of Results.  

Response: Thanks for your suggestion. We have moved section 3.1 of the original manuscript to section 2.2.2 of the revised manuscript.

Comment: Lines 32-33: Please provide references of recent studies which have been conducted and dealt with optical water properties. Please refer to (1) El Ouali, A.; El Hafyani, M.; Roubil, A.; Lahrach, A.; Essahlaoui, A.; Hamid, F.E.; Muzirafuti, A.; Paraforos, D.S.; Lanza, S.; Randazzo, G. Modeling and Spatiotemporal Mapping of Water Quality through Remote Sensing Techniques: A Case Study of the Hassan Addakhil Dam. Appl. Sci. 2021, 11, 9297. https://doi.org/10.3390/app11199297, (2) Ahmadi, B.; Gholamalifard, M.; Kutser, T.; Vignudelli, S.; Kostianoy, A. Spatio-Temporal Variability in Bio-Optical Properties of the Southern Caspian Sea: A Historic Analysis of Ocean Color Data. Remote Sens. 2020, 12, 3975. https://doi.org/10.3390/rs12233975.

Response: Thanks for your comment. We added these researches in section 1 of our revised manuscript. In detail:

Line 67. “Different optical properties will produce varied of Rrs features[36,37], which bring more challenges to the denoising.” was added.

Comment: Line 297: “The five popular spectral indices…” please provide references.

Response: Thanks for your comment. We have added references of these indices in the revised manuscript. In detail:

Line 330. “The five popular spectral indices are band ratio parameter[54,55], normalized difference chlorophyll-a index (NDCI)[56], three-band parameter[12], four-band parameter[57], and enhanced three-band parameter[58]. ”

In addition, we did some minor changes in the revised manuscript. in detail:

Minor changes:

Line 61. “algorithm” was deleted.

Line 64. “This means these methods were limited to denoise hyperspectral images.” Was added.

Line 66. “Rrs” was added.

Line 78. “collected from four routes” was added.

Line 79. “including Hongze Lake, Taihu Lake, Qiandao Lake, and Hangzhou Bay” was deleted.

Line 84. “OWT represents optical water type. The detailed description of OWTs can be found in section 2.2.1.” was added.

Reviewer 2 Report

1. It is necessary to describe the measurements in more detail in sections 2.3 and 2.4, despite the reference [37].

2. It is not clear why, when using a spectroradiometer with a range of 350-1050nm, Figure 2 shows graphs in the range of 400-800nm?

3. The purpose of placing a large number of graphs obtained from different locations in Figure 2 (d), (f), (g) is not clear. If this is done to assign them to different OWT, then it is better to do it in the form of a table.

Author Response

Comment: It is necessary to describe the measurements in more detail in sections 2.3 and 2.4, despite the reference [37].

Response: Thanks for your suggestion. We have added the measuring geometries of ASD in section 2.2.1 in our revised manuscript. In detail:

Line 112. “When the boat was anchored” was added.

Line 114. “To avoid direct solor radiation and influence of the ship, when measuring water radiance signals, the azimuth difference of ASD and solar is about 45°. And the zenith angle of ASD is also about 45°. When measuring skylight radiance, the spectroradiometer was rotated upwards by 90-120°” was added.

Comment: It is not clear why, when using a spectroradiometer with a range of 350-1050nm, Figure 2 shows graphs in the range of 400-800nm?

Response: Thanks for your comment. We didn’t show the whole graphs because the current version of the denoising method cannot handle Rrs signals of the whole spectral range. The main reason is the lack of inherent optical properties, like ahp, of the whole spectral range. The current data we can get is in the range of 400-800 nm. So, we just show spectra of this range to keep all the figures consistent and avoid misunderstanding. We added explanation about the wavelength range in our revised manuscript. In detail:

Line 137. “Note that in this research, only the Rrs curves between 400 and 800 nm were discussed. This is because of the lack of inherent optical property curves in wavelength ranges shorter than 400 nm and longer than 800nm limits bio-optical simulation.” was added.

Comment: The purpose of placing a large number of graphs obtained from different locations in Figure 2 (d), (f), (g) is not clear. If this is done to assign them to different OWT, then it is better to do it in the form of a table.

Response: Thanks for your comment. Figure 2 is used to show detailed spectra characters of Rrs in different OWTs. And it also shows general water conditions of different water body. If we simply show the OWTs in a table, the readers might be curious about the spectra shapes of our collected Rrs. So, we calculated the sample numbers in each OWTs of each sampling station and added them to our original table 1. In detail:

Line 84:

Table 1. basic information of the four cruises. OWT represents optical water type. The detailed description of OWTs can be found in section 2.2.1.

Sampling station

Hyperspectral sample number

Cchla

Taihu Lake (Aug. 1, 2019)

60 (OWT5: 3, OWT11: 37, OWT12: 20)

Hongze Lake (Nov. 12, 2020)

29 (OWT4: 1, OWT5: 14, OWT11: 13, OWT12: 2)

-

Qiandao Lake (Dec. 01, 2021)

10 (OWT2: 2, OWT3: 5, OWT9:2, OWT12: 1)

-

Hangzhou Bay (Jul. 26, 2017)

51 (OWT5: 17, OWT11: 30, OWT12: 4)

-

Despite these, we also did some minor modifications in the revised manuscript. In detail:

Line 61. “algorithm” was deleted.

Line 64. “This means these methods were limited to denoise hyperspectral images.” Was added.

Line 66. “Rrs” was added.

Line 78. “collected from four routes” was added.

Line 79. “including Hongze Lake, Taihu Lake, Qiandao Lake, and Hangzhou Bay” was deleted.

Line 84. “OWT represents optical water type. The detailed description of OWTs can be found in section 2.2.1.” was added.

Round 2

Reviewer 1 Report

no further comments